# Neural Priority Queues for Graph Neural Networks (GNNs)

## Abstract

Graph Neural Networks (GNNs) have shown considerable success in neural algorithmic reasoning. Many traditional algorithms make use of an explicit memory in the form of a data structure. However, there has been limited exploration on augmenting GNNs with external memory. In this paper, we present Neural Priority Queues, a differentiable analogue to algorithmic priority queues, for GNNs. We propose and motivate a desiderata for memory modules, and show that Neural PQs exhibit the desiderata, and reason about their use with algorithmic reasoning. This is further demonstrated by empirical results on the CLRS-30 dataset. Furthermore, we find the Neural PQs useful in capturing long-range interactions, as empirically shown on a dataset from the Long-Range Graph Benchmark.

## 1 Introduction

Algorithms and Deep Learning methods possess very fundamentally different properties. Training deep learning models to mimic algorithms would allow us to get neural models that show generalisation ability similar to the algorithms, while retaining the robustness to noise of deep learning systems. This building and training of neural networks to execute algorithmic computations is referred to as Neural Algorithmic Reasoning (Veličković & Blundell, 2021).

Architectures that align more with the underlying algorithm for the reasoning task, tend to generalize better (Xu et al., 2019). Previous works have drawn inspiration from the external memory and data structure use of programmes and algorithms, and have found success in improving the algorithmic reasoning capabilities of recurrent neural networks (RNNs) by extending them with differentiable variants for these memory and data structures (Graves et al., 2014; Grefenstette et al., 2015).

Recently, graph neural networks (GNNs) have found immense success with algorithmic tasks (Chen et al., 2020; Veličković et al., 2022). There have been works attempting to augment GNNs with memory, with majority using gates to do so. However, gated memory leads to very limited persistence. Furthermore, these works have solely focused on dynamic graphs, and extending these to non-dynamic graphs would involve significant effort.

In this paper, we propose the extension of the message passing framework of GNNs with external memory modules. We focus on adding a differentiable analogue to priority queues, as priority queues are a general data structure used by different algorithms and can be reduced to other data structures like stacks and queues. We name the thus formed framework for differentiable priority queues as 'Neural PQs'. We describe NPQ, an implementation under this framework, and also explore various variants for this.

We summarize the contributions of this paper below:

- We propose the 'Neural PQ' framework, an extension of the message-passing GNN framework to allow use of memory modules, with particular inspiration from priority queues.
- We present and motivate a set of desiderata for memory modules – (1) Memory-Persistence, (2) Permutation-Equivariance, (3) Reducibility to Priority Queues, and (4) No dependence on intermediate supervision. Past works have expressed some subsets of these as desirables.
- We propose NPQ, an implementation within this framework, that exhibits all the above mentioned properties. This is the first differentiable analogue to priority queues, and first memory module for GNNs to exhibit all the above desiderata, to the best of our knowledge.

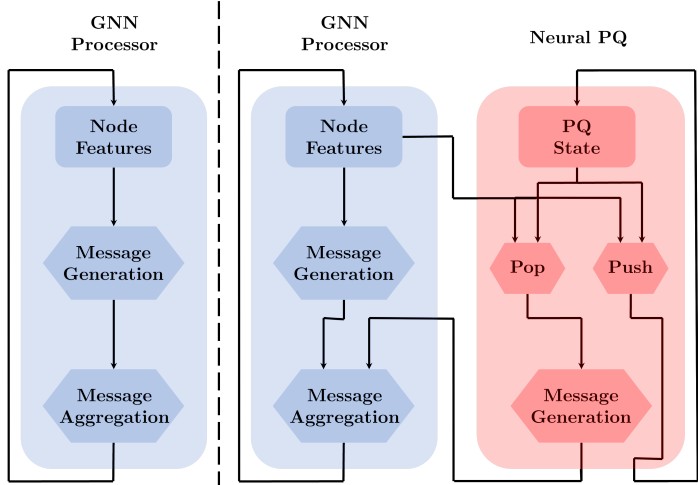

Figure 1: **Left:** A GNN processor based on the message passing framework. At each timestep, pairwise messages are formed using the node features. These messages are aggregated, and then used to update the node features for the next timestep. **Right:** The Neural PQ framework we propose. At each timestep, the node features are used to pop values from the priority queue. These values are used to form the messages that are sent to the different nodes. The node features and previous state are also used to determine what values to push, and update the priority queue.

- We perform extensive quantitative analysis, via a variety of experiments and find:
    - NPQs, when training to reason Dijkstra's shortest path algorithm, close the gap between the baseline test performance and ground truth by over $40\%$.
    - The various Neural PQs outperform the baseline on 26 out of 30 algorithms from the CLRS-30 dataset (Veličković et al., 2022). The performance gains are not restricted to algorithms that actually use a priority queue.
    - Neural PQs also help with long-range reasoning. These help local message-passing networks to capture long-range interaction. Thus, the benefits of using the Neural PQs are not limited to algorithmic reasoning, and these can be used on other tasks as well.

## 2 BACKGROUND

**CLRS Benchmark (Veličković et al., 2022)**   Various prior works have shown the efficiency of GNNs for algorithmic tasks. However, many of these works tend to be disconnected in terms of the algorithms they target, data processing and evaluation, making direct comparisons difficult. To take the first steps in solving this issue, Veličković et al. (2022) propose the CLRS Algorithmic Reasoning Benchmark which consists of 30 algorithms from the 'Introduction to Algorithms' textbook by Cormen et al. (2022). They name this dataset as CLRS-30.

The authors employ the encode-process-decode paradigm (Hamrick et al., 2018) and compare different processor networks (which are different GNNs) choices. Below we provide some more details on this encode-process-decode setup. Since we focus on the CLRS Benchmark for evaluation, this forms as the baseline architectural structure.

Let us take a graph $\mathcal{G} = (\mathcal{V}, \mathcal{E})$, with Let $\mathcal{N}_i$ as the one-hop neighbourhood of node $i$. Let $\mathbf{x}_i \in \mathbb{R}^{d_k}$ be the node features for node $i \in \mathcal{V}$, $\mathbf{e}_{ji} \in \mathbb{R}^{d_e}$ the edge features for edge $(j, i) \in \mathcal{E}$ and $\mathbf{g} \in \mathbb{R}^{d_g}$ the graph features. The *encode* step involves encoding these inputs using linear layers $f_n : \mathbb{R}^{d_k} \to \mathbb{R}^{d_h}$, $f_e : \mathbb{R}^{d_e} \to \mathbb{R}^{d_h}$ and $f_g : \mathbb{R}^{d_g} \to \mathbb{R}^{d_h}$:

$$\mathbf{h}_i = f_n(\mathbf{x}_i) \qquad \mathbf{h}_{ij} = f_e(\mathbf{e}_{ij}) \qquad \mathbf{h}_g = f_g(\mathbf{g}) \tag{1}$$

These are then used in a processor network during the *process* step. The previous latent features $\mathbf{h}_i^{(t-1)}$ are used along with the current node feature $\mathbf{h}_i$ encoding to get a recurrent encoded input $\mathbf{z}_i^{(t)}$ using a recurrent encoding function $f_A$. This recurrent cell update is line with the work of Veličković

et al. (2019). A message from node $i$ to node $j$, $\mathbf{m}_{ij}$ is computed for each pair of nodes using a message function $f_m$. These messages are aggregated using a permutation-invariant aggregation function $\bigoplus$. Finally, a readout function $f_r$ transforms the aggregated messages and node encodings into processed node latent features.

$$\mathbf{z}_i^{(t)} = f_A(\mathbf{h}_i, \mathbf{h}_i^{(t-1)}) \qquad (2) \qquad \mathbf{m}_i = \bigoplus_{j \in \mathcal{N}_i} \mathbf{m}_{ji} \qquad (4) \qquad \mathbf{h}_i^{(t)} = f_r(\mathbf{z}_i^{(t)}, \mathbf{m}_i) \quad (5)$$

$$\mathbf{m}_{ij} = f_m(\mathbf{z}_i^{(t)}, \mathbf{z}_j^{(t)}, \mathbf{h}_{ij}, \mathbf{h}_g) \ (3)$$

For different processors, $f_A$, $f_m$ and $f_r$ may differ.

The last *decode* step consists of using relevant decoding functions to get the required prediction. This might be the predicted hints or predicted output.

## 3   RELATED WORK

The ability of RNNs to work in a sequential manner led to their popularity in previous works for reasoning about algorithms, as algorithms tend to be iterative in nature. Noting that most computer programmes make use of external memory, Graves et al. (2014) proposed addition of an external memory module to RNNs, which makes reasoning about algorithms easier. Subsequent methods have worked upon this idea and have found success with memory modules inspired from different data structures. This includes Stack-Augmented RNNs by (Joulin & Mikolov, 2015) and Neural DeQues by Grefenstette et al. (2015). A key limitation of all these proposals is that they are only defined for use by RNNs. Unlike GNNs, RNNs are unable to use the structured information about the algorithms' input spaces.

Early explorations on augmenting GNNs with memory focused on the use of internal memory in the form of gates, such as Gated Graph Sequence Networks by Li et al. (2015), and Temporal Graph Networks by Rossi et al. (2020). However, the use of such RNN-like gate mechanisms limits the persistence of the graph/node histories. Persistent Message Passing (PMP) by Strathmann et al. (2021) is a noteworthy GNN that makes use of non-gated persistent external memory, by persisting some of the nodes at each timestep. However, PMPs cannot be applied to non-dynamic graphs without significant effort. Furthermore, they require intermediate-supervision.

## 4   NEURAL PQ FRAMEWORK

Previous works have proposed Neural Stacks, Queues and DeQues (Grefenstette et al., 2015; Joulin & Mikolov, 2015) that have a RNN controller. In this project, we propose the use of memory modules, with focus on differentiable PQs (or Neural PQs), with a GNN model acting as the controller. Furthermore, we propose integration of such memory modules with message-passing framework by allowing the Neural PQ to send messages to each node. The setup for this is shown in Figure 1.

### 4.1   DESIDERATA

We form the framework with the following desiderata in mind – (1) Memory-Persistence, (2) Permutation-Equivariance, (3) Reducibility to Priority Queues, and (4) No dependence on intermediate supervision. We motivate the need for these below.

Algorithms tend to run over multiple timesteps and have long temporal-interactions. **Memory-persistence** is necessary to effectively model such interactions. Furthermore, memory persistence helps avoid over-smoothing. Node embeddings of GNNs tend to start off varied, but as messages are passed, they converge to each-other, making the nodes indistinguishable. Memory-persistence would allow GNNs to remember earlier states, when the embeddings were more distinguished, and use these to promote more varied embeddings.

**Permutation-Equivariance** reflects one of the most basic symmetries of graph structures, and is necessary to ensure that isomorphic graphs receive the same representation, up to certain permutations and transformation. This makes permutation-equivariance an essential property for GNN layers.

Priority queues are a general data structure used by various algorithms. Furthermore, different data structures, like stacks and queues, can be modelled using priority queues. Since algorithmically aligned models lead to greater generalisation (Xu et al., 2019), having a memory module that **aligns with priority queues** is desired.

By **not requiring any intermediate supervision**, memory modules become easier to apply directly to different algorithmic tasks. These tasks might not use priority queues themselves, and so cannot provide the intermediate supervision.

## 4.2 FRAMEWORK

We present the framework for a Neural PQ controlled by a message-passing GNN below. We use the equations for the baseline message-passing GNN as described in Section 2. Let us suppose we have the same setup as the baseline. The encode and decode part remain the same, but now the processor uses an implementation of the Neural PQ Framework. Let the graph in consideration be $\mathcal{G} = (\mathcal{V}, \mathcal{E})$. Let the previous hidden state of the GNN be $\mathbf{h}_i^{(t-1)}$, and the previous state of the Neural PQ be $\mathbf{h}_{pq}^{(t-1)}$.

In the Neural PQ framework, we calculate the set of values $V_i$ to be popped for each node $i \in \mathcal{V}$, using a pop function $f_{pop}$. Messages are formed from these popped values using a message encoding function $f_M^{(pq)}$. Each node aggregates these messages along with the traditional node-to-node pairwise messages. Lastly, a push function $f_{push}$ updates the state of the Neural PQ, to obtain the next state $\mathbf{h}_{pq}^{(t)}$. Formally, we can define the following equations for the framework:

$$\mathbf{z}_i^{(t)} = f_A(\mathbf{h}_i, \mathbf{h}_i^{(t-1)}) \qquad (6)$$

$$\mathbf{m}_{ij} = f_m(\mathbf{z}_i^{(t)}, \mathbf{z}_j^{(t)}, \mathbf{h}_{ij}, \mathbf{h}_g) \qquad (7)$$

$$V_i = f_{pop}(\mathbf{z}_i^{(t)}, \mathbf{z}^{(t)}, \mathbf{h}_{pq}^{(t-1)}) \qquad (8)$$

$$M_i = f_M^{(pq)}(V_i, \mathbf{z}_i^{(t)}) \cup \{\mathbf{m}_{ji} \,|\, j \in \mathcal{N}_i\} \quad (9)$$

$$\mathbf{m}_i = \bigoplus_{\mathbf{m} \in M_i} \mathbf{m} \qquad (10)$$

$$\mathbf{h}_i^{(t)} = f_r(\mathbf{z}_i^{(t)}, \mathbf{m}_i) \qquad (11)$$

$$\mathbf{h}_{pq}^{(t)} = f_{push}(\mathbf{h}_{pq}^{(t-1)}, \mathbf{z}^{(t)}) \qquad (12)$$

where $\mathbf{z}^{(t)}$ is a multi-set of all the encoded inputs $\mathbf{z}_i^{(t)}$, i.e. $\mathbf{z}^{(t)} = \{\!\{\mathbf{z}_i^{(t)} \,|\, i \in \mathcal{V}\}\!\}$. $f_A, f_m$ and $f_r$ depend on which message-passing GNN processor we choose, while $f_{pop}, f_M^{(pq)}$ and $f_{push}$ depend on the Neural PQ implementation.

Note that, in the above proposed framework, we choose to delay the update of the priority queue due to pop operation until the $f_{push}$. This is done to keep the Neural PQ framework general and to segregate the queue read and update operations. This also allows us to prove the permutation-equivariance properties of the framework, as discussed below.

Even though the presented framework is inspired from priority queues, we can implement various other data structures, like queues and stacks, by appropriate $f_{pop}$ and $f_{push}$ definitions. The Neural PQ framework exhibits and promotes various properties from the desiderata. By design, these do not require any additional supervision. Furthermore, since the push, pop and message encoding functions only depend on the destination node's features, the multi-set of all node features and the Neural PQ state, all implementations are also equivariant to permutations of the nodes, under certain assumptions. For a detailed proof, refer to Appendix A.

## 5 NPQ

We propose NPQ, an implementation following the fore-mentioned Neural PQ framework that exhibits all the proposed desiderata. We divide the definition of NPQ into 4 sub-sections – (1) State, (2) Pop Function, (3) Message Encoding Function, and (4) Push Function. Taking inspiration from Neural DeQues (Grefenstette et al., 2015), NPQs consist of continuous push and pop operations.

## 5.1 STATE

Due to continuos push/pop operations, the state $\mathbf{h}_{pq}^{(t)}$ contains the values and their strengths. We can represent this as a pair of lists, one with the memory values $\mathbf{v}^{(t)} = [\mathbf{v}_1^{(t)}, \dots, \mathbf{v}_i^{(t)}, \dots]$ and the other

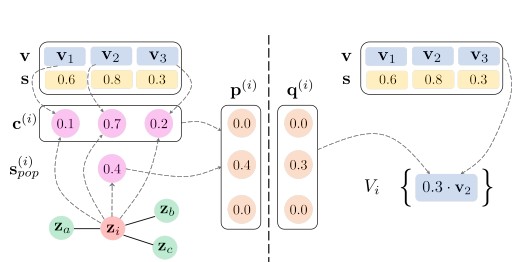

Figure 2: **State of the NPQ.** It consists of two lists of the same length, representing the values $\mathbf{v}$ in the queue and their respective strengths $\mathbf{s}$.

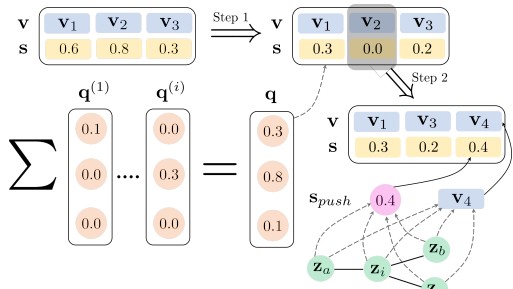

Figure 3: Sample **pop operation** for a single node $i$ in $\text{NPQ}_{\text{M}}$. **Left:** Pop-request generation. **Right:** Use of pop-grants to determine the value popped.

Figure 4: Sample **push operation**. The overall push operation can be divided into two main steps. **Step 1:** The queue is updated to reflect the removal of popped fractions. **Step 2:** New value is pushed into the queue.

with the strengths of these values $\mathbf{s}^{(t)} = [\mathbf{s}_1^{(t)}, \ldots, \mathbf{s}_i^{(t)}, \ldots]$.

$$\mathbf{h}_{pq}^{(t)} = \langle \mathbf{v}^{(t)}, \mathbf{s}^{(t)} \rangle \tag{13}$$

where $\langle \cdot \rangle$ is a tuple. Figure 2 shows a sample state for the NPQ.

### 5.2 POP FUNCTION

We propose a continuos pop function, i.e. we pop a fractional proportions of the values in the queue. This fraction, $s_{pop}^{(i)} \in (0, 1)$ for node $i$, is computed as noted in Equation 14. This equation is similar to the ones used for Neural DeQues by Grefenstette et al. (2015).

$$s_{pop}^{(i)} = \text{sigmoid}\left(f_s^{(pop)}(\mathbf{z}_i^{(t)})\right) \tag{14}$$

We use a request-grant framework to maintain the constraint that no value can be popped more than it is present, i.e. one cannot pop $0.7$ of a value $v_j$ that may be present in the PQ with only a strength of $s_j = 0.4$. Each node $i \in \mathcal{V}$ requests to pop fraction $p_j^{(i)} \in (0, 1)$ of PQ element $j$. NPQ takes all the $p_j^{(i)}$ values into consideration and grants a fraction $q_j^{(i)} \in (0, 1)$ of PQ element $j$ to node $i$, which may or may not be the same as the requested $p_j^{(i)}$. Equation 15 shows the calculation of this granted fraction given the requested fractions $p_j^{(i)}$ and the PQ element strengths $\mathbf{s}_j^{(t-1)}$.

$$q_j^{(i)} = \begin{cases} p_j^{(i)} & \text{, if } \sum_{k \in \mathcal{V}} p_j^{(k)} \leq \mathbf{s}_j^{(t-1)} \\ \frac{p_j^{(i)}}{\sum_{k \in \mathcal{V}} p_j^{(k)}} \cdot \mathbf{s}_j^{(t-1)} & \text{, else} \end{cases} \tag{15}$$

The above equation maintains the requirement that $q_j^{(i)} \leq p_j^{(i)}$ and $\sum_{i \in \mathcal{V}} q_j^{(i)} \leq \mathbf{s}_j^{(t-1)}$. These granted proportions are used to calculate the final value popped.

$$\mathbf{v} = \sum_{j \in \mathcal{I}_{pq}^{(t-1)}} q_j^{(i)} \cdot \mathbf{v}_j^{(t-1)} \tag{16} \qquad f_{pop}(\mathbf{z}_i^{(t)}, \mathbf{z}^{(t)}, \langle \mathbf{v}^{(t-1)}, \mathbf{s}^{(t-1)} \rangle) = \{\mathbf{v}\} \tag{17}$$

where $\mathcal{I}_{pq}^{(t-1)} = \left[1, \ldots, \left|\mathbf{v}^{(t-1)}\right|\right]$ is the set of indices for the NPQ. Note that since we are only popping a single value from the NPQ, we are returning a single element set. The requested pop

proportions $p_j^{(i)}$ are calculated using the continuous pop strength value $s_{pop}^{(i)}$, and attention coefficients $c_j^{(i)} \in (0, 1)$, denoting the coefficient for the $j$th element of the queue with respect to the $i$th node. These are calculated using a multi-head additive attention mechanism (Bahdanau et al., 2014; Vaswani et al., 2017). This is done with inspiration from GATs by Veličković et al. (2018).

$$e_j^{(i,h)} = \text{LeakyReLU}\left(f_{a_1}^{(h)}(\mathbf{z}_i^{(t)}) + f_{a_2}^{(h)}(\mathbf{v}_j^{(t-1)})\right) \tag{18}$$

$$\alpha_j^{(i,h)} = \text{softmax}_j\left(e_j^{(i,h)}\right) \tag{19}$$

$$c_j^{(i)} = \text{softmax}_j\left(f_a([\alpha_j^{(i,1)}, \dots, \alpha_j^{(i,h)}, \dots])\right) \tag{20}$$

where $\alpha_j^{(i,h)} \in (0, 1)$ is the attention coefficients for $j$th element of the queue with respect to node $i$ via attention-head $h$, and $f_{a_1}^{(h)}$, $f_{a_2}^{(h)}$ and $f_a$ are linear layers.

Using these coefficients, we propose two ways of popping elements from the queue – Max Popping and Weighted Popping. We refer to NPQ using max popping and weighted popping as NPQ$_M$ and NPQ$_W$, respectively. Figure 3 shows sample pop operation for NPQ.

**Max Popping**  The element $j$ of the queue with the highest attention coefficient $c_j^{(i)}$ is requested to be popped for the node $i$.

$$k = \underset{k \in \mathcal{I}_{pq}^{(t-1)}}{\arg\max}\, c_k^{(i)} \tag{21} \qquad\qquad p_j^{(i)} = s_{pop}^{(i)} \cdot \mathbb{I}_{\{k\}}(j) \tag{22}$$

where $\mathbb{I}_A(\cdot)$ is the indicator function for set $A$, i.e. $\mathbb{I}_A(a) = 1 \iff a \in A$ and $\mathbb{I}_A(a) = 0 \iff a \notin A$.

**Weighted Popping**  The attention coefficients are treated as soft-weights with which each element in the PQ is requested to be popped.

$$p_j^{(i)} = s_{pop}^{(i)} \cdot c_j^{(i)} \tag{23}$$

### 5.3 PRIORITY QUEUE MESSAGE FUNCTION

We use a simple message encoding function, where each output is passed through a linear layer $f_m^{(pq)}$.

$$f_M^{(pq)}(V_i, \mathbf{z}_i^{(t)}) = \left\{ f_m^{(pq)}(\mathbf{v}) \mid \mathbf{v} \in V_i \right\} \tag{24}$$

### 5.4 PUSH FUNCTION

As mentioned earlier, the push function is actually the state update function. Here we first delete the popped proportions from the NPQ. Let $\mathbf{h}_{pq}^{(t-1)} = \langle \mathbf{v}^{(t-1)}, \mathbf{s}^{(t-1)} \rangle$ be the previous NPQ state. Then, we can define the NPQ state with the popped proportions deleted as $\langle \mathbf{v}', \mathbf{s}' \rangle$, which are calculated as below.

$$\mathbf{s}_i' = \mathbf{s}_i^{(t-1)} - \sum_{k \in \mathcal{V}} q_i^{(k)} \tag{25} \qquad \mathbf{s}' = \text{nonzero}_i\,(\mathbf{s}_i') \tag{26} \qquad \mathbf{v}' = \mathbf{v}^{(t)}[\,\text{arg-nonzero}_i\,(\mathbf{s}_i')] \tag{27}$$

where $q_i^{(k)} \in (0, 1)$ is the proportion NPQ element $i$ granted to be popped for node $k$ as defined in Equation 15, $\text{nonzero}_i(\mathbf{s}_i')$ is sequence of $\mathbf{s}_i'$ with all zero $\mathbf{s}_i'$ removed, and similarly, arg-nonzero$_i$ is the relevant indices of the sequence.

We push a single value $\mathbf{v}$ for the whole graph. To determine this value, we pass each node embedding through a linear layer $f_v$ and sum the formed values across all the nodes. In line with Neural DeQues by Grefenstette et al. (2015), this values is activated using a tanh function to get the final value to be pushed.

The push function is continuous and so requires calculation of the push strength $\mathbf{s}_{push}$. This is done in a similar manner to the push values calculation, using a linear layer $f_s^{(push)}$. We use a logistic

sigmoid activation here instead of tanh, akin to Neural DeQues.

$$\mathbf{v} = \tanh \left( \sum_{i \in \mathcal{V}} f_v(\mathbf{z}_i^{(t)}) \right) \tag{28}$$

$$\mathbf{s} = \text{sigmoid} \left( \sum_{i \in \mathcal{V}} f_s^{(push)}(\mathbf{z}_i^{(t)}) \right) \tag{29}$$

$$f_{push}(\langle \mathbf{v}^{(t-1)}, \mathbf{s}^{(t-1)} \rangle, \mathbf{z}^{(t)}) = \langle \mathbf{v}' \,||\, [\mathbf{v}], \mathbf{s}' \,||\, [\mathbf{s}_{push}] \rangle \tag{30}$$

Note that in the above equations, we use $q_i^{(k)}$ and $\mathbf{z}_i^{(t)}$, which are not actually inputs to the $f_{push}$ function. This is done mainly to maintain readability of the functions. These equations can be easily reformulated to only use $\mathbf{h}_{pq}^{(t-1)}$ and $\mathbf{z}^{(t)}$, in order to follow the general Neural PQ framework. Refer to Appendix B for the reformulation.

## 5.5 PROPERTIES

Simply by virtue of following the Neural PQ framework, NPQ exhibits two of the desiderata – Permutation-Equivariance, and no dependence on intermediate supervision. We do not update or replace the previously stored NPQ elements, but rather persist them as long as possible, and only delete their proportions when we pop them. This allows NPQ to achieve much greater memory-persistence than done using gated memories.

Lastly, the push and pop operations of the NPQ are defined to be aligned close to the push and pop operations of the traditional priority queue. In fact, under some assumptions, we can prove that NPQ$_M$ can be reduced to a traditional priority queue. This can be done by taking the push and pop functions to be encoding the key-value pairs for the priority queue elements. For a detailed proof, refer to Appendix C.

Thus, NPQ satisfies the four stated desiderata.

## 5.6 VARIANTS

We also explore some variations on the proposed NPQ. One such variation involves consideration of greater memory-persistence by not deleting the popped elements. We refer to this variation as NPQ-P.

Notably, NPQ treats popping as a node-wise activity. We can instead treat popping as a graph operation, i.e. each node receives the same set of popped values. This can be done by either sending all the node-wise popped values to all the nodes, or by popping a single value for all the nodes. We refer to these two variants as NPQ-SA and NPQ-SV, respectively.

Empirically, we found these latter two variants more useful when combined with the first one. We refer to these combined variations as NPQ-P-SA and NPQ-P-SV, respectively. For the exact equations for the variants, refer to Appendix D-F.

## 6 EVALUATION

The main hypothesis we test is whether the Neural PQ implementations are useful for algorithmic reasoning by using the CLRS-30 dataset (Veličković et al., 2022). To do so, we undertake multiple experiments – (1) We first focus on a single algorithm, Dijkstra's Shortest Path algorithm, evaluating the performance of the Neural PQs with a MLP MPNN as the base GNN, comparing them with the MPNN baseline as well as an MLP MPNN with an oracle priority queue. (2) We also evaluate the performance of the Neural PQs on rest of the algorithms from the CLRS benchmark. (3) Lastly, we also test whether the Neural PQs are useful for long-range reasoning, by evaluating their performance on a dataset from the Long Range Graph Benchmark (Dwivedi et al., 2022). Appendix G shows some more experiments performed.

Table 1: Test performance (Mean ± Standard Deviation) of models with MLP MPNN base on learning Dijkstra's Algorithm with 256 node graphs, run with 3 different seeds. The table shows the results for the *Best* validation score model (early-stopped model) and the *Last* model in training.

| Method | Best | Last |
|---|---|---|
| Baseline | 68.58% ± 9.71 | 76.97% ± 4.38 |
| NPQ$_W$ | **85.48% ± 3.50** | **86.22% ± 2.20** |
| NPQ$_M$ | 74.54% ± 8.37 | 74.68% ± 3.06 |
| NPQ$_W$-SA | 79.74% ± 2.99 | 69.36% ± 12.02 |
| NPQ$_W$-SV | 77.04% ± 2.99 | 79.89% ± 6.28 |
| NPQ$_W$-P | 63.74% ± 11.81 | 74.74% ± 14.84 |
| NPQ$_M$-P | 73.12% ± 3.03 | 72.14% ± 6.76 |
| NPQ$_M$-P-SA | 79.19% ± 5.17 | 78.26% ± 6.19 |
| NPQ$_W$-P-SV | 71.46% ± 6.75 | 79.44% ± 5.90 |
| Oracle PQ | 75.85% ± 3.65 | 85.37% ± 3.72 |

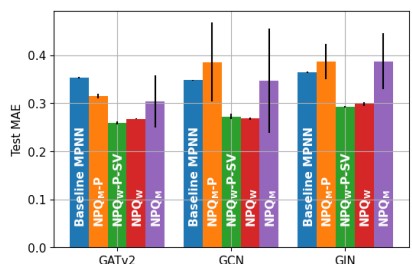

Figure 5: Test MAE (Mean ± Standard Deviation) of different Neural PQs with different base processors on Peptides-struct dataset, run with 3 different seeds. **Lower** the test MAE, **better** is the performance.

## 6.1 Dijkstra's Algorithm – MPNN Base

We train the models on Dijkstra's algorithm from CLRS-30, and test for out-of-distribution (OOD) generalisation, i.e. the models are trained on smaller input graphs, containing 16 nodes, and tested on larger graphs, containing 256 nodes. The training data consists of 1000 samples, while the testing and validation data consist of 32 samples each. We test the models on larger graph sizes than done by Veličković et al. (2022) (they use graphs with 64 nodes) to better test the generalisation ability, and because baseline MPNN model already gets around 91.5% test performance with 64 nodes.

To test the limit of attainable performance from Neural PQs, we test an MPNN with access to an Oracle PQ, where apart from the standard input features, we also take information about the values pushed and popped from the priority queue as input. The Oracle Neural PQ forces the push and pop operation to be determined by the algorithmic PQ. This information about the actual PQ is used in training, validation as well as testing.

Table 1 shows the test performance of the different models. We see that the last model performs much better than the early-stopped model for the baseline and Oracle PQ. Notably, the last and early-stopped model perform similarly for NPQ$_W$. NPQ$_W$ outperforms the baseline as well as the Oracle PQ. In fact, we see that it **closes the gap** between the test performance of baseline MPNN and true solution, i.e. 100% test performance, by over **40%**.

We also note that NPQ$_W$ outperforms the persistence variants. This suggests that the performance improvements are not simply due to introduction of persistence or a form of memory and complicated pop operations are not unnecessary. It also outperforms the graph pop operation variants, suggesting that the node-wise pop operation might be better suited for these tasks.

## 6.2 Different Algorithms from CLRS-30

We train and test five models for each algorithm from CLRS-30 dataset – 'Baseline' (no memory module), NPQ$_M$-P-SA, NPQ$_W$-P-SV, NPQ$_W$ and NPQ$_M$. We train each model on graphs with 16 nodes, and test them on graphs with 128 nodes, and consider only the early-stopped models.

Figure 6 shows the comparison for best performing Neural PQ and the baseline MPNN for each algorithm. We see that for **26 out of the 30 algorithms, at least one of the Neural PQs outperforms the baseline MPNN**. Interestingly, the optimal Neural PQ version depends on the algorithm of choice. Notably, the performance gain of using a Neural PQ does not seem to be limited to algorithms that use a traditional priority queue. This supports our belief that the Neural PQ implementations are quite general, and these can perform various roles, such as acting as a traditional data structure, or a persistent-memory for accessing past overall graph states. We provide the table with algorithm-wise performance of each Neural PQ in Appendix G. Focussing on NPQ$_M$, we found that it outperforms the baseline for 17 algorithms (more than half of the algorithms). We see that for 12

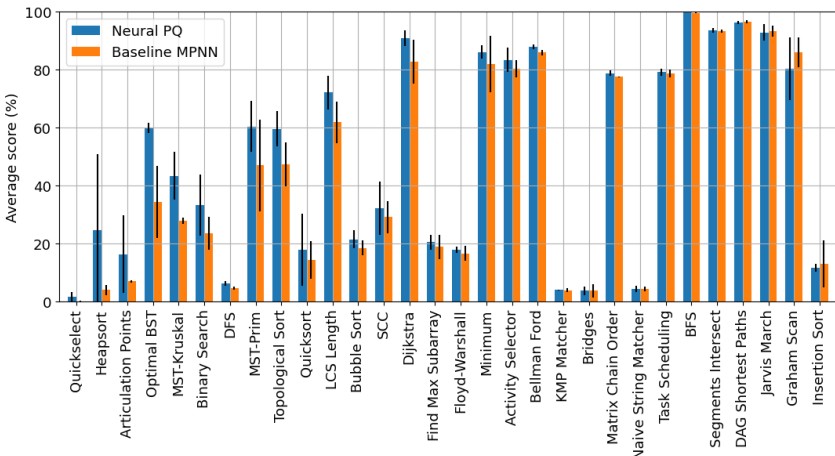

Figure 6: Evaluation results for best performing Neural PQ and the baseline MPNN model for the 30 algorithms from CLRS-30, sorted by the relative improvement in performance.

algorithms, it improves the performance or closes the gap to true prediction by at least $10\%$. For $4$ algorithms, it improves performance/reduces the gap by at least $50\%$.

### 6.3 Long-Range Reasoning

Message-passing based GNNs exchange information between 1-hop neighbours to build node representations at each layer. Past works have shown that such information propagation leads to over-squashing when the path of information traversal is long, and leading to poor performance on long-range tasks (Alon & Yahav, 2021; Dwivedi et al., 2022). Dwivedi et al. (2022) have proposed a collection of graph learning datasets to form 'Long Range Graph Benchmark' (LRGB), each requiring long-range interaction reasoning to achieve strong performance. In these experiments, we test the performance of using Neural PQs on Peptides-struct dataset from the LRGB benchmark.

Figure 5 shows the test MAE results for the different Neural PQs and the baseline. Notably, all Neural PQs outperform the baseline for GATv2 processor, while only $NPQ_W$-P-SV and $NPQ_W$ outperform the baseline on the other two processors. The success of $NPQ_W$-P-SV and $NPQ_W$ means that these Neural PQs are empirically helping the models with long-range reasoning. Notably, we see that Weighted popping seems more useful for long-range reasoning.

## 7 Conclusion and Future Works

In this paper, we proposed Neural PQs, a general framework for adding memory modules to GNNs, with inspirations from traditional priority queues. We empirically show that NPQs help with algorithmic reasoning, and without any extra supervision, match the performance of the baseline model that has access to true priority queue operations on Dijkstra's algorithm. The performance gains are not limited to algorithms using priority queues. Furthermore, we show that the Neural PQs help with capturing long-range interaction, by demonstrating their prowess on the `Peptides-struct` dataset from the Long-Range Graph Benchmark.

The success of the Neural PQs has a wide effect on the field of representational learning. It opens up a research domain exploring the use of memory modules with GNNs, especially their interfacing with the message-passing framework. The Neural PQs take crucial steps towards advancing the neural algorithmic reasoning field. These also hold potential with various other fields and tasks, as seen by their performance on the long-range reasoning task.

We have limited our focus on simple memory module operations. Potential future works could involve exploration of more complicated definitions. These might be formed by analysing the reasons behind the greater success of Neural PQs on some algorithms as opposed to others. Neural PQs can also be used for various other graph tasks, and it would be interesting to explore their uses for these.

## REPRODUCIBILITY STATEMENT

To foster reproducibility, we plan to make our code publicly available once accepted. Section 6 contains most of the experimental details, with additional details in Appendix G.

For completeness, we include further discussions and proofs in the Appendix. Appendix A contains the proof for permutation-equivariance of modules with the Neural PQ framework. To make the NPQ operations simpler, the equations defined earlier do not strictly follow the Neural PQ framework. In Appendix B, we provide a reformulation of these equations so that they follow the Neural PQ framework. We show the alignment of NPQs with traditional priority queues in Appendix C. Appendix D-F contain information and equations about the different variants of NPQ that we explored. Appendix G contains additional details and results from the experiments described earlier in the paper, as well results from further evaluations.

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

## A  PERMUTATION-EQUIVARIANCE

Node permutation-equivariance is an essential property shown by majority of the GNNs, as it embodies a key graph symmetry. A GNN layer is said to be equivariant to permutation of the nodes if and only if any permutation of the node IDs, while maintaining the overall graph structure, leads to the same permutation of the node features. Let us continue with considering our graph to be $\mathcal{G} = (\mathcal{V}, \mathcal{E})$. Let $P : \mathcal{V} \to \mathcal{V}$ be a permutation of the node IDs. For ease, let $\rho : \alpha \to \alpha$ be an overloaded permutation operation, affecting the permutation $P$ over all domains $\alpha$. For example, for the domain of vertices/nodes $\mathcal{V}$, we have $\rho(i) = P(i)$ for all $i \in \mathcal{V}$.

### A.1  MPNN PERMUTATION-EQUIVARIANCE

GNNs following the message-passing framework are permutation-equivariant. We consider the recurrent setup of CLRS benchmark here. This is fairly easy to show. First, we recall the relevant equations from Section 2 below.

$$\mathbf{z}_i^{(t)} = f_A(\mathbf{h}_i, \mathbf{h}_i^{(t-1)}) \tag{31}$$

$$\mathbf{m}_{ij} = f_m(\mathbf{z}_i^{(t)}, \mathbf{z}_j^{(t)}, \mathbf{h}_{ij}, \mathbf{h}_g) \tag{32}$$

$$\mathbf{m}_i = \bigoplus_{j \in \mathcal{N}_i} \mathbf{m}_{ji} \tag{33}$$

$$\mathbf{h}_i^{(t)} = f_r(\mathbf{z}_i^{(t)}, \mathbf{m}_i) \tag{34}$$

We can consider matrices $\mathbf{H}^{(t)}$ and $\mathbf{H}$ indexed by the vertices $i \in \mathcal{V}$, containing values $\mathbf{h}_i^{(t)}$ and $\mathbf{h}_i$, respectively. We also have a matrix of edge features $\mathbf{E}$ index by edges $(i,j) \in \mathcal{E}$ with value $\mathbf{h}_{ij}$. Further, we can define the above operations as a single layer $\mathbf{F}_{mpnn}(\cdot)$, such that:

$$\mathbf{H}^{(t)} = \mathbf{F}_{mpnn}\left(\mathbf{H}^{(t-1)}, \mathbf{H}, \mathbf{E}\right) \tag{35}$$

In order to prove that message-passing GNNs are permutation-equivariant, we need to show that:

$$\mathbf{F}_{mpnn}\left(\rho\left(\mathbf{H}^{(t-1)}\right), \rho\left(\mathbf{H}\right), \rho\left(\mathbf{E}\right)\right) = \rho\left(\mathbf{H}^{(t)}\right) \tag{36}$$

PROOF

We start by noting that by definition of $\rho$ and permutation, $\rho\left(\mathbf{H}^{(t-1)}\right)$ and $\rho\left(\mathbf{H}\right)$ are matrices such that they have values $\mathbf{h}_{\rho(i)}^{(t-1)}$ and $\mathbf{h}_{\rho(i)}$, respectively, for index $i \in \mathcal{V}$. Also, $\rho\left(\mathbf{E}\right)$ is indexed by pairs $(i,j)$, where $(\rho(i), \rho(j)) \in \mathcal{E}$, containing value $\mathbf{h}_{\rho(i)\rho(j)}$. Thus, we can define $\mathbf{H}'^{(t)}$ as below.

$$\mathbf{H}'^{(t)} = \mathbf{F}_{mpnn}\left(\rho\left(\mathbf{H}^{(t-1)}\right), \rho\left(\mathbf{H}\right), \rho\left(\mathbf{E}\right)\right) \tag{37}$$

where $\mathbf{H}'^{(t)}$ has value $\mathbf{h}_i'^{(t)}$ for index $i \in \mathcal{V}$, with $\mathbf{h}_i'^{(t)}$ as defined below.

$$\mathbf{z}_i'^{(t)} = f_A(\mathbf{h}_{\rho(i)}, \mathbf{h}_{\rho(i)}^{(t-1)}) \tag{38}$$

$$\mathbf{m}_{ij}' = f_m(\mathbf{z}_i'^{(t)}, \mathbf{z}_j'^{(t)}, \mathbf{h}_{\rho(i)\rho(j)}, \mathbf{h}_g) \tag{39}$$

$$\mathbf{m}_i' = \bigoplus_{j \in \rho(\mathcal{N}_{\rho(i)})} \mathbf{m}_{ji}' \tag{40}$$

$$\mathbf{h}_i'^{(t)} = f_r(\mathbf{z}_i'^{(t)}, \mathbf{m}_i') \tag{41}$$

where $\rho(\mathcal{N}_{\rho(i)})$ is the one-hop neighbourhood on the permuted graph, and can be simply defined as $\rho(\mathcal{N}_{\rho(i)}) = \{j \in \mathcal{V} \mid (\rho(j), \rho(i)) \in \mathcal{E}\}$. All these equations follow simply from application of the MPNN equations, as noted before, on the permuted matrices.

We start by noting that $f_A(\mathbf{h}_{\rho(i)}, \mathbf{h}_{\rho(i)}^{(t-1)})$ is simply the value $\mathbf{z}_{\rho(i)}^{(t)}$. Thus, we get the below equation.

$$\mathbf{z}_i'^{(t)} = \mathbf{z}_{\rho(i)}^{(t)} \tag{42}$$

Using this in equation 39, we get:

$$\mathbf{m}'_{ij} = f_m(\mathbf{z}^{(t)}_{\rho(i)}, \mathbf{z}^{(t)}_{\rho(j)}, \mathbf{h}_{\rho(i)\rho(j)}, \mathbf{h}_g) \tag{43}$$
$$= \mathbf{m}_{\rho(i)\rho(j)} \tag{44}$$

Using this in equation 40, we get:

$$\mathbf{m}'_i = \bigoplus_{j \in \rho(\mathcal{N}_{\rho(i)})} \mathbf{m}_{\rho(j)\rho(i)} \tag{45}$$

We also note that, by definition of $\rho(\mathcal{N}_{\rho(i)})$, we get the following.

$$j \in \rho(\mathcal{N}_{\rho(i)}) \iff \rho(j) \in \mathcal{N}_{\rho(i)} \tag{46}$$

Using this in Equation 45, we get:

$$\mathbf{m}'_i = \bigoplus_{\rho(j) \in \mathcal{N}_{\rho(i)}} \mathbf{m}_{\rho(j)\rho(i)} \tag{47}$$
$$= \mathbf{m}_{\rho(i)} \tag{48}$$

Substituting the above value and value from Equation 42 in Equation 41:

$$\mathbf{h}'^{(t)}_i = f_r(\mathbf{z}^{(t)}_{\rho(i)}, \mathbf{m}_{\rho(i)}) \tag{49}$$
$$= \mathbf{h}^{(t)}_{\rho(i)} \tag{50}$$

But that means that $\mathbf{H}'^{(t)} = \rho(\mathbf{H}^{(t)})$.

$$\therefore \mathbf{F}_{mpnn}\left(\rho\left(\mathbf{H}^{(t-1)}\right), \rho\left(\mathbf{H}\right), \rho\left(\mathbf{E}\right)\right) = \rho\left(\mathbf{H}^{(t)}\right) \tag{51}$$

Hence, proved that message-passing GNNs show node-permutation equivariance.

## A.2 NEURAL PQ PERMUTATION-EQUIVARIANCE

In a similar vein, we can show that the memory modules following the Neural PQ framework proposed by me, show node-permutation equivariance. Below we recall the equations for the Neural PQ framework.

$$\mathbf{z}^{(t)}_i = f_A(\mathbf{h}_i, \mathbf{h}^{(t-1)}_i) \tag{52}$$
$$\mathbf{m}_{ij} = f_m(\mathbf{z}^{(t)}_i, \mathbf{z}^{(t)}_j, \mathbf{h}_{ij}, \mathbf{h}_g) \tag{53}$$
$$V_i = f_{pop}(\mathbf{z}^{(t)}_i, \mathbf{z}^{(t)}, \mathbf{h}^{(t-1)}_{pq}) \tag{54}$$
$$M_i = f^{(pq)}_M(V_i, \mathbf{z}^{(t)}_i) \cup \{\mathbf{m}_{ji} \,|\, j \in \mathcal{N}_i\} \tag{55}$$
$$\mathbf{m}_i = \bigoplus_{\mathbf{m} \in M_i} \mathbf{m} \tag{56}$$
$$\mathbf{h}^{(t)}_i = f_r(\mathbf{z}^{(t)}_i, \mathbf{m}_i) \tag{57}$$
$$\mathbf{h}^{(t)}_{pq} = f_{push}(\mathbf{h}^{(t-1)}_{pq}, \mathbf{z}^{(t)}) \tag{58}$$

where $\mathbf{z}^{(t)}$ is a multi-set of all the encoded inputs $\mathbf{z}^{(t)}_i$, i.e. $\mathbf{z}^{(t)} = \{\{\mathbf{z}^{(t)}_i \,|\, i \in \mathcal{V}\}\}$.

We can take $\mathbf{H}^{(t)}$, and $\mathbf{E}$ as defined in the previous section. Then, we can define the overall operations of the Neural PQ as a single layer $\mathbf{F}_{npq}(\cdot)$, such that:

$$\mathbf{H}^{(t)}, \mathbf{h}^{(t)}_{pq} = \mathbf{F}_{npq}\left(\mathbf{H}^{(t-1)}, \mathbf{H}, \mathbf{E}, \mathbf{h}^{(t-1)}_{pq}\right) \tag{59}$$

In order to prove that modules following the Neural PQ framework are permutation-equivariant, we need to show that:

$$\mathbf{F}_{npq}\left(\rho\left(\mathbf{H}^{(t-1)}\right), \rho\left(\mathbf{H}\right), \rho\left(\mathbf{E}\right), \mathbf{h}^{(t-1)}_{pq}\right) = \rho\left(\mathbf{H}^{(t)}\right), \mathbf{h}^{(t)}_{pq} \tag{60}$$

PROOF

The description of $\rho\left(\mathbf{H}^{(t-1)}\right)$, $\rho\left(\mathbf{H}\right)$ and $\rho\left(\mathbf{E}\right)$ follow here same as before. We can define $\mathbf{H}'^{(t)}$ and $\mathbf{h}'^{(t)}_{pq}$ as below.

$$\mathbf{H}'^{(t)}, \mathbf{h}'^{(t)}_{pq} = \mathbf{F}_{npq}\left(\rho\left(\mathbf{H}^{(t-1)}\right), \rho\left(\mathbf{H}\right), \rho\left(\mathbf{E}\right), \mathbf{h}^{(t-1)}_{pq}\right) \tag{61}$$

Thus, $\mathbf{H}'^{(t)}$ has value $\mathbf{h}'^{(t)}_i$ for index $i \in \mathcal{V}$, with $\mathbf{h}'^{(t)}_i$ and $\mathbf{h}'^{(t)}_{pq}$ as defined below.

$$\mathbf{z}'^{(t)}_i = f_A(\mathbf{h}_{\rho(i)}, \mathbf{h}^{(t-1)}_{\rho(i)}) \tag{62}$$

$$\mathbf{m}'_{ij} = f_m(\mathbf{z}'^{(t)}_i, \mathbf{z}'^{(t)}_j, \mathbf{h}_{\rho(i)\rho(j)}, \mathbf{h}_g) \tag{63}$$

$$V'_i = f_{pop}(\mathbf{z}'^{(t)}_i, \mathbf{z}'^{(t)}, \mathbf{h}^{(t-1)}_{pq}) \tag{64}$$

$$M'_i = f^{(pq)}_M(V'_i, \mathbf{z}'^{(t)}_i) \cup \{\mathbf{m}'_{ji} \mid j \in \rho(\mathcal{N}_{\rho(i)})\} \tag{65}$$

$$\mathbf{m}'_i = \bigoplus_{\mathbf{m} \in M'_i} \mathbf{m} \tag{66}$$

$$\mathbf{h}'^{(t)}_i = f_r(\mathbf{z}'^{(t)}_i, \mathbf{m}'_i) \tag{67}$$

$$\mathbf{h}'^{(t)}_{pq} = f_{push}(\mathbf{h}^{(t-1)}_{pq}, \mathbf{z}'^{(t)}) \tag{68}$$

where $\mathbf{z}'^{(t)}$ is a multi-set of all the node embeddings $\mathbf{z}'^{(t)}_i$, i.e. $\mathbf{z}'^{(t)} = \{\{\mathbf{z}'^{(t)}_i \mid i \in \mathcal{V}\}\}$.

The following can be shown in a similar fashion as the previous proof:

$$\mathbf{z}'^{(t)}_i = \mathbf{z}^{(t)}_{\rho(i)} \tag{69}$$

$$\mathbf{m}'_{ij} = \mathbf{m}_{\rho(i)\rho(j)} \tag{70}$$

$$j \in \rho(\mathcal{N}_{\rho(i)}) \iff \rho(j) \in \mathcal{N}_{\rho(i)} \tag{71}$$

Using Equation 69 and the definition of $\mathbf{z}'^{(t)}$, we get:

$$\mathbf{z}'^{(t)} = \{\{\mathbf{z}^{(t)}_{\rho(i)} \mid i \in \mathcal{V}\}\} \tag{72}$$

$$= \{\{\mathbf{z}^{(t)}_i \mid i \in \mathcal{V}\}\} \tag{73}$$

$$= \mathbf{z}^{(t)} \tag{74}$$

because $\rho(i) = P(i)$ and $P$ is a permutation, and so the multi-sets are equal.

Using Equation 69 and Equation 74, we can update Equation 64 as below.

$$V'_i = f_{pop}(\mathbf{z}^{(t)}_{\rho(i)}, \mathbf{z}^{(t)}, \mathbf{h}^{(t-1)}_{pq}) \tag{75}$$

$$= V_{\rho(i)} \tag{76}$$

Substituting Equation 69, Equation 70 and Equation 76 in Equation 65, we get:

$$M'_i = f^{(pq)}_M(V_{\rho(i)}, \mathbf{z}^{(t)}_{\rho(i)}) \cup \{\mathbf{m}_{\rho(i)\rho(j)} \mid j \in \rho(\mathcal{N}_{\rho(i)})\} \tag{77}$$

Using Equation 71 in the above equation, we get:

$$M'_i = f^{(pq)}_M(V_{\rho(i)}, \mathbf{z}^{(t)}_{\rho(i)}) \cup \{\mathbf{m}_{\rho(i)\rho(j)} \mid \rho(j) \in \mathcal{N}_{\rho(i)}\} \tag{78}$$

$$= M_{\rho(i)} \tag{79}$$

Substituting this in Equation 66, we get:

$$\mathbf{m}'_i = \bigoplus_{\mathbf{m} \in M_{\rho(i)}} \mathbf{m} \tag{80}$$

$$= \mathbf{m}_{\rho(i)} \tag{81}$$

Using this and Equation 69 in Equation 67, we get:

$$\mathbf{h}'^{(t)}_i = f_r(\mathbf{z}^{(t)}_{\rho(i)}, \mathbf{m}_{\rho(i)}) \tag{82}$$

$$= \mathbf{h}^{(t)}_{\rho(i)} \tag{83}$$

This means that $\mathbf{H}'^{(t)} = \rho(\mathbf{H}^{(t)})$.

Additionally, substituting the value from Equation 74 in Equation 68, we get:

$$\mathbf{h}'^{(t)}_{pq} = f_{push}(\mathbf{h}^{(t-1)}_{pq}, \mathbf{z}^{(t)}) \tag{84}$$

$$= \mathbf{h}^{(t)}_{pq} \tag{85}$$

Since, $\mathbf{H}'^{(t)} = \rho(\mathbf{H}^{(t)})$ and $\mathbf{h}'^{(t)}_{pq} = \mathbf{h}^{(t)}_{pq}$, we have:

$$\mathbf{F}_{npq}\left(\rho\left(\mathbf{H}^{(t-1)}\right), \rho\left(\mathbf{H}\right), \rho\left(\mathbf{E}\right), \mathbf{h}^{(t-1)}_{pq}\right) = \rho\left(\mathbf{H}^{(t)}\right), \mathbf{h}^{(t)}_{pq} \tag{86}$$

Hence, proved, that modules following the Neural PQ framework show node-permutation equivariance.

## B  NPQ REFORMULATION

In Section 5, we introduced the push and pop operations for NPQ. However, the equations defined there make use of the granted pop proportions $q^{(i)}_j$ (and $\mathbf{z}^{(t)}_i$ as well in the push operation). These are not exactly available to the respective functions as defined in the Neural PQ framework. However, these are used in Section 5 only to make the equations easier to understand, and they instead can be reformulated to conform to the Neural PQ framework. We provide the reformulation below.

### B.1  POP FUNCTION

The calculation of pop request fractions $p^{(i)}_j$ as defined in Section 5.2 can be combined into a single function POP-REQUEST, which takes $\mathbf{z}^{(t)}_i$, $\mathbf{h}^{(t-1)}_{pq}$ and $j$ – the embedding of node $i$, previous NPQ state and the index of the queue element we want to calculate the pop request for, and returns the pop request fraction $p^{(i)}_j$.

$$p^{(i)}_j = \text{POP-REQUEST}(\mathbf{z}^{(t)}_i, \mathbf{h}^{(t-1)}_{pq}, j) \tag{87}$$

We can calculate the sum of the pop requests as below:

$$\text{TOT-POP-REQ}(\mathbf{z}^{(t)}, \mathbf{h}^{(t-1)}_{pq}, j) = \sum_{\mathbf{z}_k \in \mathbf{z}^{(t)}} \text{POP-REQUEST}(\mathbf{z}_k, \mathbf{h}^{(t-1)}_{pq}, j) \tag{88}$$

By definition of $\mathbf{z}^{(t)}$, we have:

$$\sum_{k \in \mathcal{V}} p^{(k)}_j = \sum_{k \in \mathcal{V}} \text{POP-REQUEST}(\mathbf{z}^{(t)}_k, \mathbf{h}^{(t-1)}_{pq}, j) \tag{89}$$

$$= \sum_{\mathbf{z}_k \in \mathbf{z}^{(t)}} \text{POP-REQUEST}(\mathbf{z}_k, \mathbf{h}^{(t-1)}_{pq}, j) \tag{90}$$

$$= \text{TOT-POP-REQ}(\mathbf{z}^{(t)}, \mathbf{h}^{(t-1)}_{pq}, j) \tag{91}$$

Using this, we can reformulate the pop proportions $q^{(i)}_j$ as below:

$$q^{(i)}_j = \begin{cases} p^{(i)}_j & \text{, if TOT-POP-REQ}(\mathbf{z}^{(t)}, \mathbf{h}^{(t-1)}_{pq}, j) \leq \mathbf{s}^{(t-1)}_j \\ \frac{p^{(i)}_j}{\text{TOT-POP-REQ}(\mathbf{z}^{(t)}, \mathbf{h}^{(t-1)}_{pq}), j} \cdot \mathbf{s}^{(t-1)}_j & \text{, else} \end{cases} \tag{92}$$

It is easy to see that this reformulation conforms to the pop function as defined in the Neural PQ framework.

## B.2 PUSH FUNCTION

We can rewrite the push function as below:

$$\mathbf{s}'_i = \mathbf{s}_i^{(t-1)} - \min\left(\text{TOT-POP-REQ}(\mathbf{z}^{(t)}, \mathbf{h}_{pq}^{(t-1)}, i), \mathbf{s}_i^{(t-1)}\right) \quad (93)$$

$$\mathbf{s}' = \text{nonzero}_i\left(\mathbf{s}'_i\right) \quad (94)$$

$$\mathbf{v}' = \mathbf{v}^{(t)}\left[\text{arg-nonzero}_i\left(\mathbf{s}'_i\right)\right] \quad (95)$$

$$\mathbf{v} = \tanh\left(\sum_{\mathbf{z}_k \in \mathbf{z}^{(t)}} f_v(\mathbf{z}_k)\right) \quad (96)$$

$$\mathbf{s} = \text{sigmoid}\left(\sum_{\mathbf{z}_k \in \mathbf{z}^{(t)}} f_s^{(push)}(\mathbf{z}_k)\right) \quad (97)$$

$$f_{push}(\langle \mathbf{v}^{(t-1)}, \mathbf{s}^{(t-1)}\rangle, \mathbf{z}^{(t)}) = \langle \mathbf{v}' \,||\, [\mathbf{v}], \mathbf{s}' \,||\, [\mathbf{s}_{push}]\rangle \quad (98)$$

Again, it is easy to see that the above reformulation conforms to the push function as defined in the Neural PQ framework.

## C PRIORITY QUEUE ALIGNMENT

Following is the priority queue setup we consider. We will then show that under certain assumptions, we can reduce the NPQ$_\text{M}$ computation to the equations for the priority queue setup defined.

Let us suppose some algorithm uses a priority queue. We shall take the algorithm to push at most 1 element and pop at most 1 element in each timestep. We take the pushing and popping to be controlled by the overall graph, but under certain assumptions, the reduction can be extended to having these from nodes instead. Further, we take that the output of the popping is returned to some specific node. Let $P^{(t-1)} = \{(k_1, \nu_1), \ldots, (k_i, \nu_i), \ldots\}$ be the set of past un-popped key-value-pair pushes to the priority queue. Let $\mathbf{o}_i^{(t)}$ be the output to the node $i \in \mathcal{V}$. We further assume that all priority keys and values are unique. We can represent the operation of a traditional priority queue over a timestep as below, using $P^{(t-1)}$ from the previous timestep and by calculating the next $P^{(t-1)}$ and the outputs $\mathbf{o}_i^{(t)}$. We take the value returned to be $\mathbf{0}$ if no value is returned to the node.

$$\mathbf{o}_i^{(t)} = \begin{cases} \nu_{max}^{(t-1)} & \text{, if we pop this timestep and return to node } i \\ \mathbf{0} & \text{, else} \end{cases} \quad (99)$$

$$P'^{(t)} = \begin{cases} P^{(t-1)} - (k_{max}^{(t-1)}, \nu_{max}^{(t-1)}) & \text{, if we pop this timestep} \\ P^{(t-1)} & \text{, else} \end{cases} \quad (100)$$

$$P^{(t)} = \begin{cases} P'^{(t)} \cup (k^{(t)}, \nu^{(t)}) & \text{, if we push some key-value pair } (k^{(t)}, \nu^{(t)}) \\ P'^{(t)} & \text{, else} \end{cases} \quad (101)$$

where $(k_{max}^{(t-1)}, \nu_{max}^{(t-1)}) \in P^{(t-1)}$ such that $\forall (k, \nu) \in P^{(t-1)}. \ k \le k_{max}$.

We shall now show that, under certain assumptions, the NPQ$_\text{M}$ operations can be reduced to the above operations. More specifically, we shall show that the NPQ state $\mathbf{h}_{pq}^{(t)}$ mimics the priority queue state $P^{(t)}$, and the NPQ messages $M_i$ mimics the returned value $\mathbf{o}_i^{(t)}$. The main assumptions we make is that the linear layers are capable of expressing the required functions and the intermediate embedding sizes are big enough to not lose any information. We go into more details about the assumptions as we describe the reduction.

We continue with the graph $\mathcal{G} = (\mathcal{V}, \mathcal{E})$ setup, with $\mathbf{z}_i^{(t)}$ as the node features. Since NPQ uses a GNN controller, we assume that we can make all decisions from the node features, i.e. the node features determine whether we want to push and pop values and if so, what value and key to push, and which node to pop to. Let $\mathbf{h}_{pq}^{(t-1)} = \langle \mathbf{v}^{(t-1)}, \mathbf{s}^{(t-1)}\rangle$ be the previous NPQ state, such that each element of $\mathbf{v}^{(t-1)}$ is an encoding of a unique key-value-pair in $P^{(t-1)}$, with an element existing for each key-value pair. Let $\kappa$ be the mapping from NPQ values to the corresponding keys, and $\omega$ be the mapping from NPQ to the values in $P^{(t-1)}$. Let for all $s \in \mathbf{s}^{(t-1)}, s = 1$.

**Pop Function**   We shall now breakdown the pop function, to make the overall computation match the traditional priority queue's. We assume that we can instantiate $f_s^{(pop)}$ in a manner such that $s_{pop}^{(i)} = 1$ iff we want to pop a value for node $i$ in timestep $t$, else $s_{pop}^{(i)} = 0$. Let us further suppose that the attentional mechanism calculating the coefficient $c_j^{(i)}$ simply extracts the encoded priority key in $\mathbf{v}_j^{t-1}$. More specifically, coefficient $c_j^{(i)}$ is calculated as below in NPQ.

$$e_j^{(i,h)} = \text{LeakyReLU}\left(f_{a_1}^{(h)}(\mathbf{z}_i^{(t)}) + f_{a_2}^{(h)}(\mathbf{v}_j^{(t-1)})\right) \tag{102}$$

$$\alpha_j^{(i,h)} = \text{softmax}_j\left(e_j^{(i,h)}\right) \tag{103}$$

$$c_j^{(i)} = \text{softmax}_j\left(f_a([\alpha_j^{(i,1)}, \ldots, \alpha_j^{(i,h)}, \ldots])\right) \tag{104}$$

For simplicity, we can take number of attention heads to be 1. Let $f_{a_1}^{(h)}(\mathbf{x}) = \mathbf{0}$ for all $\mathbf{x}$. Further, suppose that $f_{a_2}^{(h)}(\mathbf{v}_j^{(t-1)}) = \text{LeakyReLU}^{-1}(\kappa(\mathbf{v}_j^{(t-1)}))$. Also, let us take $f_a$ to be an identity function. Thus, we get the below equation for $c_j^{(i)}$.

$$e_j^{(i,h)} = \kappa(\mathbf{v}_j^{(t-1)}) \tag{105}$$

$$\alpha_j^{(i,h)} = \text{softmax}_j\left(\kappa(\mathbf{v}_j^{(t-1)})\right) \tag{106}$$

$$c_j^{(i)} = \text{softmax}_j\left(\text{softmax}_j\left(\kappa(\mathbf{v}_j^{(t-1)})\right)\right) \tag{107}$$

Since we use Max Popping, we have the pop proportions requested as below.

$$k = \underset{k \in \mathcal{I}_{pq}^{(t-1)}}{\text{argmax}} \, \text{softmax}_k\left(\text{softmax}_k\left(\kappa(\mathbf{v}_k^{(t-1)})\right)\right) \tag{108}$$

$$p_j^{(i)} = s_{pop}^{(i)} \cdot \mathbb{I}_{\{k\}}(j) \tag{109}$$

We can show easily that $\text{argmax}_{a \in A} \, \text{softmax}_a(b_a) = \text{argmax}_{a \in A} \, b_a$, for some set $A$ and values $b_a$. Thus, we can simplify the pop proportions.

$$k = \underset{k \in \mathcal{I}_{pq}^{(t-1)}}{\text{argmax}} \, \kappa(\mathbf{v}_k^{(t-1)}) \tag{110}$$

$$p_j^{(i)} = s_{pop}^{(i)} \cdot \mathbb{I}_{\{k\}}(j) \tag{111}$$

But $\text{argmax}_{k \in \mathcal{I}_{pq}^{(t-1)}} \kappa(\mathbf{v}_k^{(t-1)})$ is nothing but index of the NPQ element corresponding to $k_{max}^{(t-1)}$. Thus, we can re-formulate the above equation to use this.

$$p_j^{(i)} = s_{pop}^{(i)} \cdot \mathbb{I}_{\left\{k_{max}^{(t-1)}\right\}}\left(\kappa(\mathbf{v}_j^{(t-1)})\right) \tag{112}$$

Using the assumption about $f_s^{(pop)}$ and rewriting the indicator function, we get the following equation.

$$p_j^{(i)} = \begin{cases} 1 & \text{, if we pop this timestep, return to node } i \\ & \quad \text{and } \kappa(\mathbf{v}_j^{(t-1)}) = k_{max}^{(t-1)} \\ 0 & \text{, else} \end{cases} \tag{113}$$

Since $\forall s \in \mathbf{s}^{(t-1)} . \, s = 1$, NPQ will fully grant each pop request. Thus, we have $q_j^{(i)} = p_j^{(i)}$. NPQ has pop function's output values as defined below.

$$f_{pop}(\mathbf{z}_i^{(t)}, \mathbf{z}^{(t)}, \langle \mathbf{v}^{(t-1)}, \mathbf{s}^{(t-1)} \rangle) = \left\{ \sum_{j \in \mathcal{I}_{pq}^{(t-1)}} q_j^{(i)} \cdot \mathbf{v}_j^{(t-1)} \right\} \tag{114}$$

Using the previous equations, we get the following reduction for $V_i$.

$$V_i = \begin{cases} \{\mathbf{v}_j^{(t-1)}\} & \text{, if we pop this timestep and return to node } i \\ & \quad \text{where } \kappa(\mathbf{v}_j^{(t-1)}) = k_{max}^{(t-1)} \\ \{\mathbf{0}\} & \text{, else} \end{cases} \tag{115}$$

**Message Encoding Function**  We assume that NPQ learns a message encoding function $f_m^{(pq)}$ such that $f_m^{(pq)}(\mathbf{0}) = \mathbf{0}$ and $f_m^{(pq)}(\mathbf{v}_j^{(t-1)}) = \omega(\mathbf{v}_j^{(t-1)})$ for all $j$. Thus, the reduction for $M_i$ is as follows.

$$M_i = \begin{cases} \nu_{max}^{(t-1)} & \text{, if we pop this timestep and return to node } i \\ \mathbf{0} & \text{, else} \end{cases} \tag{116}$$

where we make use of the fact that $\kappa(\mathbf{v}_j^{(t-1)}) = k_{max}^{(t-1)} \iff \omega(\mathbf{v}_j^{(t-1)}) = \nu_{max}^{(t-1)}$ which follows by the definition of $\omega$, $\kappa$ and the key-value-pair $(k_{max}^{(t-1)}, \nu_{max}^{(t-1)})$.

This in fact is the same value as the output value $\mathbf{o}_i^{(t)}$, returned in the traditional priority queue. Thus, we have shown that NPQ messages mimic the returned output. We only need to now show that the state update can be mimicked as well.

**Push Function**  It is straightforward to see, albeit somewhat tedious to show, that the NPQ state $\langle \mathbf{v}', \mathbf{s}' \rangle$ is such that $\forall s \in \mathbf{s}'. s = 1$ and that the correspondence between $\mathbf{v}'$ and the key-value-pairs in $P'^{(t)}$ is maintained by $\kappa$ and $\omega$. Thus, we use this directly without proof.

Similar to the push strength, we assume that we can instantiate $f_s^{(push)}$ in a manner such that $\mathbf{s} = 1$ iff we want to push a value in timestep $t$, else $\mathbf{s} = 0$. We also assume that we can instantiate $f_v$ such that when we want to push a value, we get $\mathbf{v}$ as a unique recoverable encoding of the key-value-pair that we want to push. That means that we can now define another mapping, $\kappa'$ and $\omega'$, such that these are equal to $\kappa$ and $\omega$ for the previous elements, and for the new element, these are equal to the newly pushed key-value-pair. This means that the new state NPQ $\mathbf{h}_{pq}^{(t)}$ mimics the priority queue state $P^{(t)}$.

Hence proved that we can reduce the operations of NPQ$_M$ to a traditional priority queues, under the noted assumptions.

## D  GREATER MEMORY-PERSISTENCE – NPQ-P

NPQ defined earlier deletes the popped elements from the queue. In this variation, we explore a more persistent Neural PQ implementation. We refer to this as NPQ-Persistent or NPQ-P. NPQ-P does not delete popped elements. Here the pop operation acts more like a 'seek' operation, i.e. it just returns the element but does not delete it. We make a further simplification by making the push and pop operations discrete. Thus, at each timestep, one element is pushed into the queue, and one element (or a weighted combination of elements) per node is read and passed as a message to each node. Below we note the changes in the components for this implementation, as compared to NPQ. The message encoding function remains the same, but the rest change.

### D.1  STATE

Since we do not have a continuous push and pop operation, we do not need to keep track of the strengths of the different values in the queue. Thus, the state of the priority queue $\mathbf{h}_{pq}^{(t)}$ is simply the list of memory values. For the sake of consistency with NPQ, we represent the state as a single value tuple.

$$\mathbf{h}_{pq}^{(t)} = \langle \mathbf{v}^{(t)} \rangle \tag{117}$$

### D.2  POP FUNCTION

As noted, pop does not delete an element from the queue. The priority of the elements of the queue is determined using attention coefficients $c_j^{(i)} \in (0, 1)$, denoting the coefficient for the $j$th element of the queue with respect to the $i$th node, as defined in Equation 20. Since we no longer have push and pop strengths, we no longer need the request-grant framework. We again have two popping strategies – Max Popping and Weighted Popping, and related implementations NPQ$_M$-P and NPQ$_W$-P respectively.

MAX POPPING

$$j = \operatorname*{argmax}_{j \in \mathcal{I}_{pq}^{(t-1)}} c_j^{(i)} \tag{118}$$

$$f_{pop}(\mathbf{z}_i^{(t)}, \mathbf{z}^{(t)}, \langle \mathbf{v}^{(t-1)} \rangle) = \left\{ \mathbf{v}_j^{(t-1)} \right\} \tag{119}$$

WEIGHTED POPPING

$$f_{pop}(\mathbf{z}_i^{(t)}, \mathbf{z}^{(t)}, \langle \mathbf{v}^{(t-1)} \rangle) = \left\{ \sum_{j \in \mathcal{I}_{pq}^{(t-1)}} c_j^{(i)} \cdot \mathbf{v}_j^{(t-1)} \right\} \tag{120}$$

### D.3 PUSH FUNCTION

Since we are no longer deleting elements, the push function simply consists of appending new push value to the queue.

$$\mathbf{v} = \tanh\left( \sum_{i \in \mathcal{V}} f_v(\mathbf{z}_i^{(t)}) \right) \tag{121}$$

$$f_{push}(\langle \mathbf{v}^{(t-1)} \rangle, \mathbf{z}^{(t)}) = \langle \mathbf{v}^{(t-1)} \, || \, [\mathbf{v}] \rangle \tag{122}$$

## E   GRAPH PRIORITY QUEUE – NPQ-SA

In NPQ, each node pops different elements from the queue, and thus receives different messages from the queue. This node-wise treatment of the priority queue might not be always ideal, and we might want to treat the priority queue messaging to be uniform for the whole graph, i.e. we might want each node to receive the same values from the Neural PQ. In NPQ Send to All or NPQ-SA, we propose a variant that returns the same set of popped values for each node. This set is simply the union of all the values that would have been popped from the queue in NPQ for the different nodes. Thus, in NPQ-SA, each node receives $|\mathcal{V}|$ values from the queue. All the components of the Neural PQ remain the same as in Section 5 except for the pop function, which is as follows.

### E.1   POP FUNCTION

Most of the function remains the same and uses the attention coefficients and pop fractions $q_j^{(i)}$ as defined in Section 5.4. The changes are as below, where we now first calculate $V_k'$, the set of values popped for node $k$, if we were using the pop function of NPQ. These are then union-ed to get the values returned for each node.

$$V_k' = \left\{ \sum_{j \in \mathcal{I}_{pq}^{(t-1)}} q_j^{(k)} \cdot \mathbf{v}_j^{(t-1)} \right\} \tag{123}$$

$$f_{pop}(\mathbf{z}_i^{(t)}, \mathbf{z}^{(t)}, \langle \mathbf{v}^{(t-1)} \rangle) = \bigcup_{k \in \mathcal{V}} V_k' \tag{124}$$

## F   GRAPH PRIORITY QUEUE – NPQ-SV

NPQ-SA is one way to model a graph controlled priority queue. Another way would be to only pop a single value from the priority queue, and return this single value to all nodes. We implement this strategy in NPQ Single Value or NPQ-SV. Again, only the pop function changes here.

### F.1 POP FUNCTION

NPQ-SV makes two changes to the pop function of NPQ – (1) all pop strengths are aggregated to get a single value for all the nodes, and (2) all the attention coefficients are aggregated to get the same values for all the nodes. We calculate this single pop strength $s_{pop}$ and attention coefficients $c_j$ as below.

$$s_{pop} = \text{ sigmoid}\left(\sum_{i \in \mathcal{V}} f_s^{(pop)}(\mathbf{z}_i^{(t)})\right) \quad (125)$$

$$c_j = \text{ softmax}_j \left(\sum_{i \in \mathcal{V}} c_j^{(i)}\right) \quad (126)$$

where $c_j^{(i)}$ is the node-wise attention coefficients, as calculated in Section 5.4. We now need to just update the pop requests $p_j^{(i)}$ to use these values, as done below. Rest of the function remains the same as in NPQ.

#### MAX POPPING

$$k = \operatorname*{argmax}_{k \in \mathcal{I}_{pq}^{(t-1)}} c_k \quad (127)$$

$$p_j^{(i)} = s_{pop} \cdot \mathbb{I}_{\{k\}}(j) \quad (128)$$

#### WEIGHTED POPPING

$$p_j^{(i)} = s_{pop} \cdot c_j \quad (129)$$

## G    FURTHER EVALUATIONS

In Section 6, we present various experiments we performed. In this section, we provide further details about the experiments and their results, along with some more experiments.

### G.1    DIJKSTRA'S ALGORITHM – DIFFERENT BASE PROCESSORS

We run experiments to test whether the performance improvements seen with the MPNN controlling the Neural PQs are also seen with the use of different processors controlling the Neural PQs. We continue with Dijkstra's algorithm as the target task, and explore the use of different base processors – Deep Sets (Zaheer et al., 2018), GAT (Veličković et al., 2018), GATv2 (Brody et al., 2022), PGN (Veličković et al., 2020) and Triplet MPNN (Ibarz et al., 2022), apart from MPNN. We compare the performance of baseline (no memory module), NPQ$_M$-P-SA, NPQ$_W$-P-SV, NPQ$_W$ and NPQ$_M$, each with these different base processors.

Table 2 shows the test performance of the different Neural PQs when used with the above mentioned different base processors/controllers, on graphs with 256 nodes. We observe that for each processor, at least one of the Neural PQs outperforms the baseline. NPQ$_W$ outperforms the baseline for all processors, except Deep Sets and PGN, for both of which, it gets a performance very close to the baseline. Thus, the use of Neural PQs does not seem to be limited to the basic MLP MPNN, although interestingly, for some processor GNNs, like Deep Sets and PGN here, other Neural PQ variants seem to be more useful.

### G.2    DIFFERENT ALGORITHMS FROM CLRS

Section 6.2 talks about our experiment of using the different Neural PQs for all 30 algorithms from CLRS-30. Table 3 shows the algorithm-wise performance for each model. This also shows the Win/Tie/Loss counts, which are calculated in the same manner as Veličković et al. (2022). Table 4 shows the algorithm-wise win/tie/loss of each model.

Table 2: Test performance (Mean ± Standard Deviation) of different Neural PQs with different base processors on learning Dijkstra's Algorithm, run with 3 different seeds. Tested on graphs with 256 nodes, and with Early-stopped model.

| Processor | Baseline | $\text{NPQ}_\text{M}$-P-SA | $\text{NPQ}_\text{W}$-P-SV | $\text{NPQ}_\text{W}$ | $\text{NPQ}_\text{M}$ |
|---|---|---|---|---|---|
| Deep Sets | $31.01\% \pm 2.41$ | $\mathbf{37.76}\% \pm \mathbf{0.60}$ | $31.30\% \pm 3.12$ | $30.36\% \pm 3.47$ | $31.04\% \pm 5.50$ |
| GAT | $36.94\% \pm 12.51$ | $19.58\% \pm 9.17$ | $\mathbf{47.04}\% \pm \mathbf{13.51}$ | $38.72\% \pm 9.13$ | $36.96\% \pm 14.37$ |
| GATv2 | $55.62\% \pm 3.18$ | $19.07\% \pm 12.56$ | $52.12\% \pm 13.29$ | $\mathbf{58.99}\% \pm \mathbf{12.02}$ | $57.01\% \pm 14.89$ |
| MPNN | $76.97\% \pm 4.38$ | $81.64\% \pm 3.71$ | $77.19\% \pm 8.15$ | $\mathbf{86.22}\% \pm \mathbf{2.20}$ | $74.68\% \pm 3.06$ |
| PGN | $65.28\% \pm 6.16$ | $67.58\% \pm 7.01$ | $\mathbf{74.42}\% \pm \mathbf{5.50}$ | $64.57\% \pm 3.00$ | $59.58\% \pm 9.46$ |
| Triplet MPNN | $57.65\% \pm 7.43$ | $66.58\% \pm 19.72$ | $74.31\% \pm 8.68$ | $\mathbf{79.92}\% \pm \mathbf{11.49}$ | $73.92\% \pm 6.18$ |

Table 3: Test performance (Mean ± Standard Deviation) of the Neural PQs on out-of-distribution test data for all 30 algorithms from CLRS-30, run with 3 different seeds.

| Algorithm | Baseline | $\text{NPQ}_\text{M}$-P-SA | $\text{NPQ}_\text{W}$-P-SV | $\text{NPQ}_\text{W}$ | $\text{NPQ}_\text{M}$ |
|---|---|---|---|---|---|
| Activity Selector | $80.38\% \pm 2.94$ | $81.29\% \pm 1.35$ | $62.27\% \pm 19.30$ | $78.82\% \pm 6.13$ | $\mathbf{83.36}\% \pm \mathbf{4.27}$ |
| Articulation Points | $6.86\% \pm 0.33$ | $10.27\% \pm 2.16$ | $\mathbf{16.30}\% \pm \mathbf{13.49}$ | $9.51\% \pm 2.37$ | $13.40\% \pm 2.96$ |
| Bellman Ford | $85.90\% \pm 0.94$ | $\mathbf{87.92}\% \pm \mathbf{0.86}$ | $80.43\% \pm 3.45$ | $86.91\% \pm 0.69$ | $87.50\% \pm 2.03$ |
| BFS | $99.57\% \pm 0.16$ | $99.85\% \pm 0.08$ | $\mathbf{99.86}\% \pm \mathbf{0.12}$ | $99.84\% \pm 0.09$ | $99.64\% \pm 0.18$ |
| Binary Search | $23.53\% \pm 5.76$ | $19.21\% \pm 7.90$ | $\mathbf{33.28}\% \pm \mathbf{10.49}$ | $21.22\% \pm 10.96$ | $28.79\% \pm 2.37$ |
| Bridges | $3.61\% \pm 2.39$ | $\mathbf{3.67}\% \pm \mathbf{1.45}$ | $1.68\% \pm 0.72$ | $2.46\% \pm 1.32$ | $1.83\% \pm 0.40$ |
| Bubble Sort | $18.44\% \pm 2.52$ | $14.31\% \pm 8.79$ | $16.11\% \pm 7.30$ | $\mathbf{21.37}\% \pm \mathbf{3.11}$ | $17.78\% \pm 10.43$ |
| DAG Shortest Paths | $\mathbf{96.48}\% \pm \mathbf{0.49}$ | $95.64\% \pm 1.16$ | $69.96\% \pm 21.89$ | $94.93\% \pm 1.79$ | $96.19\% \pm 0.60$ |
| DFS | $4.48\% \pm 0.57$ | $\mathbf{6.16}\% \pm \mathbf{0.88}$ | $5.30\% \pm 0.91$ | $4.17\% \pm 0.92$ | $6.03\% \pm 1.54$ |
| Dijkstra | $82.69\% \pm 7.50$ | $89.72\% \pm 0.73$ | $89.48\% \pm 2.24$ | $\mathbf{90.93}\% \pm \mathbf{2.66}$ | $83.94\% \pm 2.52$ |
| Find Max Subarray | $18.77\% \pm 4.28$ | $\mathbf{20.38}\% \pm \mathbf{2.66}$ | $12.39\% \pm 3.86$ | $16.05\% \pm 3.58$ | $19.94\% \pm 5.31$ |
| Floyd-Warshall | $16.53\% \pm 2.49$ | $15.57\% \pm 3.86$ | $10.49\% \pm 6.39$ | $13.50\% \pm 4.87$ | $\mathbf{17.84}\% \pm \mathbf{1.09}$ |
| Graham Scan | $\mathbf{85.92}\% \pm \mathbf{5.10}$ | $80.33\% \pm 10.77$ | $67.95\% \pm 3.09$ | $64.05\% \pm 15.28$ | $61.95\% \pm 20.81$ |
| Heapsort | $3.99\% \pm 1.76$ | $\mathbf{24.58}\% \pm \mathbf{26.33}$ | $10.57\% \pm 4.91$ | $10.45\% \pm 6.30$ | $16.37\% \pm 8.33$ |
| Insertion Sort | $\mathbf{12.99}\% \pm \mathbf{8.14}$ | $11.61\% \pm 1.45$ | $10.02\% \pm 2.96$ | $9.88\% \pm 1.31$ | $8.16\% \pm 1.22$ |
| Jarvis March | $\mathbf{93.35}\% \pm \mathbf{1.91}$ | $80.35\% \pm 20.57$ | $73.55\% \pm 6.28$ | $89.03\% \pm 4.10$ | $92.88\% \pm 2.87$ |
| KMP Matcher | $3.82\% \pm 0.63$ | $3.12\% \pm 0.54$ | $\mathbf{3.91}\% \pm \mathbf{0.15}$ | $2.59\% \pm 1.30$ | $3.23\% \pm 0.82$ |
| LCS Length | $61.84\% \pm 7.18$ | $\mathbf{72.05}\% \pm \mathbf{5.72}$ | $64.21\% \pm 2.29$ | $59.36\% \pm 2.24$ | $67.50\% \pm 6.24$ |
| Matrix Chain Order | $77.50\% \pm 0.21$ | $73.48\% \pm 0.16$ | $75.45\% \pm 2.03$ | $76.90\% \pm 2.01$ | $\mathbf{78.74}\% \pm \mathbf{1.01}$ |
| Minimum | $81.99\% \pm 9.79$ | $79.61\% \pm 19.99$ | $76.86\% \pm 18.03$ | $\mathbf{86.08}\% \pm \mathbf{2.36}$ | $75.65\% \pm 14.35$ |
| MST-Kruskal | $27.84\% \pm 1.03$ | $29.53\% \pm 15.24$ | $34.80\% \pm 21.05$ | $27.56\% \pm 5.96$ | $\mathbf{43.24}\% \pm \mathbf{8.25}$ |
| MST-Prim | $46.88\% \pm 15.85$ | $52.59\% \pm 8.58$ | $46.28\% \pm 10.93$ | $49.04\% \pm 8.70$ | $\mathbf{60.31}\% \pm \mathbf{8.79}$ |
| Naïve String Matcher | $4.21\% \pm 0.87$ | $\mathbf{4.24}\% \pm \mathbf{0.98}$ | $2.82\% \pm 1.96$ | $3.83\% \pm 0.25$ | $3.40\% \pm 1.55$ |
| Optimal BST | $34.27\% \pm 12.39$ | $26.83\% \pm 9.77$ | $22.91\% \pm 25.26$ | $18.66\% \pm 15.76$ | $\mathbf{59.89}\% \pm \mathbf{1.84}$ |
| Quickselect | $0.05\% \pm 0.07$ | $0.02\% \pm 0.03$ | $0.74\% \pm 0.81$ | $\mathbf{1.48}\% \pm \mathbf{1.67}$ | $0.00\% \pm 0.00$ |
| Quicksort | $14.20\% \pm 6.53$ | $15.42\% \pm 7.90$ | $10.70\% \pm 3.06$ | $17.07\% \pm 8.36$ | $\mathbf{17.71}\% \pm \mathbf{12.42}$ |
| Segments Intersect | $93.27\% \pm 0.56$ | $\mathbf{93.53}\% \pm \mathbf{0.88}$ | $93.44\% \pm 0.83$ | $93.08\% \pm 0.31$ | $93.19\% \pm 0.06$ |
| SCC | $29.12\% \pm 5.56$ | $\mathbf{32.19}\% \pm \mathbf{9.23}$ | $21.22\% \pm 6.05$ | $29.86\% \pm 3.44$ | $22.50\% \pm 2.18$ |
| Task Scheduling | $78.66\% \pm 1.26$ | $\mathbf{79.10}\% \pm \mathbf{1.23}$ | $78.52\% \pm 0.75$ | $78.96\% \pm 0.58$ | $77.79\% \pm 0.26$ |
| Topological Sort | $47.23\% \pm 7.58$ | $\mathbf{59.54}\% \pm \mathbf{6.13}$ | $55.11\% \pm 0.21$ | $52.87\% \pm 0.52$ | $50.98\% \pm 2.69$ |
| Overall Average | $44.48\%$ | $45.40\%$ | $41.55\%$ | $43.68\%$ | $\mathbf{46.32}\%$ |
| Win/Tie/Loss Counts | 2/16/12 | 0/18/12 | 0/12/18 | 2/12/16 | **4/15/11** |

Table 4: Win/Tie/Loss counts of the Neural PQs and Baseline on out-of-distribution test data for all 30 algorithms from CLRS-30, run with 3 different seeds.

| Algorithm | Baseline | NPQ$_M$-P-SA | NPQ$_W$-P-SV | NPQ$_W$ | NPQ$_M$ |
|---|---|---|---|---|---|
| Activity Selector | *T* | *T* | L | L | *T* |
| Articulation Points | *T* | *T* | *T* | *T* | *T* |
| Bellman Ford | L | *T* | L | L | *T* |
| BFS | L | *T* | *T* | *T* | L |
| Binary Search | *T* | L | *T* | L | *T* |
| Bridges | *T* | *T* | L | *T* | L |
| Bubble Sort | *T* | L | L | *T* | L |
| DAG Shortest Paths | *T* | L | L | L | *T* |
| DFS | L | *T* | *T* | L | *T* |
| Dijkstra | L | *T* | *T* | *T* | L |
| Find Max Subarray | *T* | *T* | L | L | *T* |
| Floyd-Warshall | L | L | L | L | **W** |
| Graham Scan | **W** | L | L | L | L |
| Heapsort | *T* | *T* | *T* | *T* | *T* |
| Insertion Sort | *T* | *T* | *T* | *T* | *T* |
| Jarvis March | *T* | L | L | L | *T* |
| KMP Matcher | *T* | L | *T* | L | L |
| LCS Length | L | *T* | L | L | *T* |
| Matrix Chain Order | L | L | L | L | **W** |
| Minimum | L | L | L | **W** | L |
| MST-Kruskal | L | L | L | L | **W** |
| MST-Prim | L | *T* | L | L | *T* |
| Naïve String Matcher | *T* | *T* | L | *T* | *T* |
| Optimal BST | L | L | L | L | **W** |
| Quickselect | *T* | *T* | *T* | *T* | *T* |
| Quicksort | *T* | *T* | *T* | *T* | *T* |
| Segments Intersect | *T* | *T* | *T* | *T* | *T* |
| SCC | *T* | *T* | L | *T* | L |
| Task Scheduling | *T* | *T* | *T* | *T* | L |
| Topological Sort | L | *T* | *T* | L | L |
| Overall Counts | 1/17/12 | 0/19/11 | 0/13/17 | 1/13/16 | **4/16/10** |

## G.3  LONG-RANGE REASONING

Table 5 shows the evaluation results for the baseline and Neural PQs on the Peptides-Struct dataset from Long-Range Graph Benchmark (Dwivedi et al., 2022) for different processors, as detailed in Section 6.3.

Table 5: Test MAE (Mean $\pm$ Standard Deviation) of different Neural PQs with different base processors on Peptides-struct dataset, run with 3 different seeds. **Lower** the test MAE, **better** is the performance.

| Processor | Baseline | NPQ$_M$-P | NPQ$_W$-P-SV | NPQ$_W$ | NPQ$_M$ |
|---|---|---|---|---|---|
| GATv2 | $0.3530 \pm 0.0019$ | $0.3141 \pm 0.0049$ | $\mathbf{0.2589 \pm 0.0031}$ | $0.2670 \pm 0.0009$ | $0.3037 \pm 0.0540$ |
| GCN | $0.3476 \pm 0.0003$ | $0.3854 \pm 0.0825$ | $0.2723 \pm 0.0054$ | $\mathbf{0.2678 \pm 0.0023}$ | $0.3462 \pm 0.1088$ |
| GINE | $0.3640 \pm 0.0010$ | $0.3865 \pm 0.0372$ | $\mathbf{0.2922 \pm 0.0012}$ | $0.2984 \pm 0.0043$ | $0.3871 \pm 0.0578$ |

