# OpenReview forum: "Neural Priority Queues for Graph Neural Networks (GNNs)"
_ICLR.cc/2024/Conference — ICLR 2024 Conference Withdrawn Submission_

### Official Review · Reviewer_1rbd · 2023-10-30

**Soundness:** 2 fair
**Presentation:** 2 fair
**Contribution:** 3 good
**Rating:** 3
**Confidence:** 4

**Summary:**

The paper deals with GNN memory extensions, mainly for the purpose of algorithmic learning, but also extends it to one real-world benchmark. The data structure that is introduced is supposed to mimic a priority queue that all nodes concurrently interface with. Notably, all operations are implemented in a differentiable manner, making it possible to learn end-to-end.

**Strengths:**

The paper makes use of differentiable data structures in GNNs. The results seem promising for the purpose of algorithmic reasoning, and there is evidence that it could also be helpful for other tasks.

**Weaknesses:**

First, the major weaknesses

1. Poor reproducibility
The paper does provide barely any information that would make it possible to implement the proposed architecture. While mathematical formulas are provided, there is considerable room for interpretation regarding the actual implementation. These include, but are not limited to: Where are linear layers applied, and what uses MLPs? What is the baseline GNN architecture? How are the models trained (there is absolutely zero information on that)? What are the model sizes (especially for the LRGB dataset, is the parameter limit adhered to)? What hardware was used, and how long did the training take? To be clear, these are just some exemplary questions, the authors should provide enough information to make their results as reproducible as possible. Sadly, this makes the empirical results questionable as they cannot be verified. No code was provided, which would have largely alleviated this problem.

2. The main claim of alignment with priority queues is not well-supported
	The paper claims that the architecture aligns with neural priority queues (at least this is the implication):
“In this paper, we present Neural Priority Queues, a differentiable analogue to algorithmic priority queues, for GNNs.”
I don’t see why this is the case. I understand that the architecture can be reduced to priority queues, but this does not mean that it aligns with them. In fact, the supposed priority queue has no notion of “priority”. Nodes attend to a set of values and choose which one they want to pop. A real priority queue only pops the element with the highest priority, which is not enforced or incentivized here. The reduction only claims that one could learn to store both keys and values in v, and then attend to the element with the highest priority, without any evidence that the learning dynamics even allow for this. Overall, just because the data structure is expressive enough to mimic a priority queue in theory, it doesn’t mean that it is one. Indeed, one could take any data store and let nodes attend to it. To support the claim that this is a neural priority queue, the authors should test if it can learn the behavior of one, potentially even without the whole GNN around it. For example, an analysis of attention coefficients and what elements in the queue are popped would help here. Even then, it is debatable if this should be called a “neural priority queue”.

3. Weak contextualization, missing related work, and almost no good baselines in the evaluation.
The paper is missing highly relevant related architectures, and the related work section does not provide much information. Some claims need a citation, and others are, at the very least, misleading. For example:
“However, gated memory leads to very limited persistence. Furthermore, these works have solely focused on dynamic graphs, and extending these to non-dynamic graphs would involve significant effort.”
The first sentence is not backed up, and the second hints at the claim that memory-augmented GNNs have only been developed for dynamic graphs, but what about works like: https://memari-workshop.github.io/papers/paper_32.pdf ?
Here is a whole survey on memory-augmented architectures for GNNs with many more examples (albeit viewed through the lens of neuroscience): https://arxiv.org/pdf/2209.10818.pdf
While the missing related work and their citations are concerning on their own, this also has an impact on the empirical evaluation, where none of these baselines are used in the comparison. The fact that some other architectures use gating does not provide a good reason not to compare to them. Further, the experiments could also contextualize the results better by providing architectures that allow to attend to values of other nodes (e.g., graph transformers or virtual nodes that represent storage). As it stands, there is only a comparison between “neural PQs” and the same GNN without them (with maybe the exception of the oracle PQ). Whether the performance boost is due to the smart usage of some data structure is unclear. On another note, the work seems to build heavily on the work of Grefenstette et al., which introduces differentiable data structures such as deques, stacks, and queues. Before introducing a new variation of these data structures (the “priority” queue), it would make sense to use the existing ones in the GNN framework to further show that the proposed data structure works better for neural algorithmic reasoning. In fact, exactly this has already been done before (https://www.researchgate.net/profile/Fpj_Nijweide/publication/365623739_Differentiable_Data_Structures_for_Neural_Algorithmic_Reasoning/links/637b9e4537878b3e87ccdff9/Differentiable-Data-Structures-for-Neural-Algorithmic-Reasoning.pdf), so why is it missing here?

Minor weaknesses

4. Non-standard evaluation for CLRS
The authors claim that they diverge from the graph sizes used in CLRS for testing (256 instead of 64) “to better test the generalisation ability, and because baseline MPNN model already gets around 91.5% test performance with 64 nodes.”
I don’t see why an accuracy of 91.5% should be considered too good for testing, and other algorithms in CLRS are far from this accuracy. The generalization abilities are never mentioned in the evaluation after this, and this does not justify why no results on the standard test graph size of 64 are provided. This makes me curious: Do “neural PQs” not perform well for small graphs and only generalize better?
Further, for the evaluation of all 30 algorithms, a graph size of 128 is used (without any justification). Is there actually a reason for this? The authors should probably provide performance scores on graphs of sizes 64, 128, and 256.

5. Claims about long-range information exchange
The authors claim that because their models perform well on one (!) dataset from LRGB, this has to be due to long-range information exchange. However, a recent paper (https://www.researchgate.net/publication/373642031_Where_Did_the_Gap_Go_Reassessing_the_Long-Range_Graph_Benchmark) has shown that, although being called long-range graph benchmark, there is no evidence to support that these problems actually require long-range information. This makes these claims problematic. Further, the authors only use a single dataset from LRGB.

6. The paper is sometimes hard to follow
The authors use many mathematical formulas with even more variables and briefly mention what some of these do, but the descriptions are hard to follow. Some variables or functions are barely introduced/explained. Figure 4 shows the push operation, but elements are popped first. This is also confusing to me.
Moreover, there are many variations of the “neural PQ” that are not motivated or explained (at least in the main paper). A thorough evaluation of the different techniques is not provided.
Further, the “Oracle PQ” is not introduced very well, for example, this sentence is not clear to me:
“The Oracle Neural PQ forces the push and pop operation to be determined by the algorithmic PQ”
What is the “algorithmic PQ”?

7. Desiderata not clear
The authors do not mention what they understand under “Memory Persistence”, the first desiderata. This is not a standard terminology in the community, and a definition/criterium when this “memory persistence” is fulfilled is required. At the moment, the authors only claim that it holds for their architecture, all without mentioning what it is. Desiderata 3 (Reducibility to Priority Queues) kind of hints at the fact that the architecture is not necessarily aligned with priority queues (as mentioned before).

In summary, this paper presents an interesting idea with promising results, but due to the aforementioned reasons, I cannot vouch to accept it. The work is not reproducible, the main claim of a differentiable priority queue is not well supported, and there is almost no mention of relevant related work. The other weaknesses only add to the image of a paper that needs to be considerably improved until it is ready for acceptance. Of course, I’m happy to discuss all points with the authors and to increase my score if the weaknesses can be fixed. Until then, my score is a reject.

**Questions:**

In addition to the questions raised by the weaknesses above, I have the following:
1. What about the scalability and theoretical complexity of the approach? Can you give some runtime bounds? Are there challenges when applying the architecture to larger graphs?
2. How many of the underlying algorithms in CLRS-30 actually make use of a priority queue? Is it just Dijkstra? Why do others perform better or worse then?
3. What are the performances on graph size 64?
4. For the early-stopping discussion (Table 1), what is the size of graphs in the validation set?

---

### Official Review · Reviewer_t69k · 2023-10-31

**Soundness:** 2 fair
**Presentation:** 2 fair
**Contribution:** 2 fair
**Rating:** 3
**Confidence:** 3

**Summary:**

The paper proposes a neural Priority Queue (PQ), as a differentiable analogue of algorithmic PQ. The proposed neural PQ is equipped with a memory module. Experimental studies show the performance of neural PQ on CLRS benchmark.

**Strengths:**

S1. The paper proposes a neural network to mimic algorithmic PQ, where the neural architecture has a memory module.

S2. Experiments are conducted on a wide range of algorithms which needs PQ.

**Weaknesses:**

W1. The motivation of why we need a neural PQ is unclear.

W1.1 (Effectiveness) In many algorithms, PQ is used for best-first search or heuristic search.  However, as a neural network, the neural PQ may not always correctly mimic each push/pop operation of algorithmic. It’s unclear to which degree, the neural PQ can be an effective alternative of algorithmic PQ, regarding fulfilling corresponding best-first/heuristic search.

W1.2 (Efficiency) The paper does not present any complexity analysis. It seems compared with algorithmic PQ, the complexity of neural PQ cannot have many advantages.

The reviewer understands the authors want to design neural data structures. However, the research motivation is questionable. Compared with algorithmic data structures, neural data structures do not have their advantages in practice, especially for classical algorithm design.

W2. There is only one baseline, which makes the experiments less solid.

W3. For long-range reasoning, the authors only compared with a baseline model MPNN. If the authors want to emphasize the contribution of neural PQ on reasoning, they should compare more approaches that are specifically designed for reasoning.

**Questions:**

Q1. As the paper mention, the proposed neural PQ can be used for graph reasoning. What’s the difference between conducting reasoning by neural PQ and existing differentiable reasoning approaches (e.g., Neural LP, RNNLogic, Query2Box )?

[NIPS’17] Differentiable Learning of Logical Rules for Knowledge Base Reasoning

[ICLR’21] Rnnlogic: Learning Logic Rules For Reasoning On Knowledge Graphs

[ICLR’20] Query2box: Reasoning over knowledge graphs in vector space using box embedding

Q2. What’s the evaluation metric of Table 1, 2, 3? And why does that metric reflect the effectiveness of the neural data structures?

---

### Official Review · Reviewer_mXBj · 2023-10-31

**Soundness:** 3 good
**Presentation:** 2 fair
**Contribution:** 2 fair
**Rating:** 5
**Confidence:** 3

**Summary:**

This paper introduces a method named NPQ, which extends the message-passing framework by incorporating external memory modules.

**Strengths:**

The exploration of the neural algorithmic reasoning task using memory modules is an innovative and intriguing approach.

**Weaknesses:**

1. The explanation and benefits of the memory module are unclear.
The paper utilizes the memory module for reasoning the graph algorithm, but it lacks a comprehensive introduction of the memory modules, making it difficult to understand.

2. The evaluation of the proposed method is somewhat lacking.
The evaluation of the NPQ method is only conducted using the baseline MPNN. Generally, existing methods like [1] have proposed simulating the graph algorithms bellman-ford. Therefore, when evaluating the effectiveness of the proposed method, these existing methods should also be taken into account.




[1] Neural bellman-ford networks: A general graph neural network framework for link prediction.

**Questions:**

Please check the weakness

---

### Official Review · Reviewer_rwJQ · 2023-11-03

**Soundness:** 3 good
**Presentation:** 3 good
**Contribution:** 3 good
**Rating:** 6
**Confidence:** 3

**Summary:**

This paper focuses on extending GNNs with external memory for algorithmic reasoning. The authors propose a Neural Priority Queue framework that retains message passing GNNs and has flexible designs for priority queues. They also discuss several desirable property for the memory modules, which motivates their specific designs. The experiments for algorithmic reasoning on CLRS-30 dataset show the superiority of their method.

**Strengths:**

1. Overall, the paper is well-written and clear. The motivation for adding memory modules for GNNs is also sound.
2. The empirical experiments seem comprehensive. Especially for Dijkstra's algorithm, it is good to see the experiments are extended to larger graphs compared with previously used graphs with 64 nodes. This result can help us better understand the performance gap.
3. It's interesting to see the author also have some experiments about long-range reasoning. To me, the potential advantage of adding memory modules is indeed to help capture long-range information. Although, I think it might be better to have 1 or 2 more datasets on long-range reasoning.

**Weaknesses:**

1. Some equations and descriptions in Sec 5.2 are a bit confusing. I understand that $s_{pop}^{(i)}$ in Eq (14) is the proportion to pop the values in the queue. The requested fraction $p_j^{(i)}$ is calculated based on $s_{pop}^{(i)}$ and the granted fraction $q$ is based on $p$. In this sense, I would suggest somehow re-arrange the equations to make the flow more clear. For example, move Eq (14) to right before Eq (22) or move the specific pop Eq (22) and (23) to right after Eq (14). Otherwise, it's a bit confusing how to get $p$ and how it relates to Eq (15) at the first glance.
2. For the limitation of requiring intermediate supervision in previous work, it's not very clear. Could you explain more in detail and also elaborate more on how your framework can avoid this? In the current draft, I didn't quite get it fully by reading just a few lines on the top of Page 4.

**Questions:**

1. In Sec 6.1, the authors show that Oracle PQ performs worse than the proposed method. I am curious what the reason is for that. In my view, the oracle PQ may contain most of the information required to be successful in Dijkstra's algorithm, and by adding an MPPNP with access to the Oracle PR, it should provide very good performance. Please correct me if my understanding is not true. And I would appreciate it if you could provide more insights regarding this. I feel it could be more interesting than the only numerical results.